# Machine Learning-Based Prediction of Glioma *IDH* Gene Mutation Status Using Physio-Metabolic MRI of Oxygen Metabolism and Neovascularization (A Bicenter Study)

**DOI:** 10.3390/cancers16061102

**Published:** 2024-03-08

**Authors:** Andreas Stadlbauer, Katarina Nikolic, Stefan Oberndorfer, Franz Marhold, Thomas M. Kinfe, Anke Meyer-Bäse, Diana Alina Bistrian, Oliver Schnell, Arnd Doerfler

**Affiliations:** 1Karl Landsteiner University of Health Sciences, 3500 Krems, Austria; katarina.nikolic@stpoelten.lknoe.at (K.N.); stefan.oberndorfer@stpoelten.lknoe.at (S.O.); franz.marhold@stpoelten.lknoe.at (F.M.); 2Institute of Medical Radiology, Diagnostics, Intervention, University Hospital St. Pölten, 3100 St. Pölten, Austria; 3Department of Neurosurgery, Universitätsklinikum Erlangen, Friedrich-Alexander University (FAU) Erlangen-Nürnberg, 91054 Erlangen, Germany; thomasmehari.kinfe@uk-erlangen.de (T.M.K.); oliver.schnell@uk-erlangen.de (O.S.); 4Division of Neurology, University Hospital St. Pölten, 3100 St. Pölten, Austria; 5Division of Neurosurgery, University Hospital St. Pölten, 3100 St. Pölten, Austria; 6Division of Functional Neurosurgery and Stereotaxy, Friedrich-Alexander University (FAU) Erlangen-Nürnberg, 91054 Erlangen, Germany; 7Department of Scientific Computing, Florida State University, 400 Dirac Science Library Tallahassee, Tallahassee, FL 32306-4120, USA; ameyerbaese@fsu.edu; 8Department of Electrical Engineering and Industrial Informatics, Politehnica University of Timisoara, 300006 Timișoara, Romania; diana.bistrian@fih.upt.ro; 9Department of Neuroradiology, Universitätsklinikum Erlangen, Friedrich-Alexander University (FAU) Erlangen-Nürnberg, 91054 Erlangen, Germany; arnd.doerfler@uk-erlangen.de

**Keywords:** artificial intelligence, deep learning, glioma, isocitrate dehydrogenase gene, *IDH* gene mutation, neurooncology, machine learning, physio-metabolic MRI, preoperative classification

## Abstract

**Simple Summary:**

Early characterization of the isocitrate dehydrogenase (*IDHIDH*) gene mutation status of glioma is crucial for personalized decision making and prognosis in clinical neurooncological treatment. Based on the known differences in energy metabolism between *IDHIDH*-mutated and *IDHIDH*-wildtype gliomas, we assessed physio-metabolic magnetic resonance imaging-based measures along with machine learning for potential reliable presurgical characterization of *IDHIDH* gene status. Traditional machine learning algorithms and simple deep learning models trained in analyzing physio-metabolic parameters demonstrated the best performance in classifying the *IDHIDH* gene status of gliomas in independent internal testing. In contrast, external testing revealed that traditional machine learning models trained on clinical MRI data had higher accuracy compared to physio-metabolic algorithms, reflecting differences in data acquisition methodology between the two sites. Our results outline the necessity of independent internal and external testing, thus calling for standardized and robust protocols for clinical MRI-driven AI applications.

**Abstract:**

The mutational status of the isocitrate dehydrogenase (*IDH*) gene plays a key role in the treatment of glioma patients because it is known to affect energy metabolism pathways relevant to glioma. Physio-metabolic magnetic resonance imaging (MRI) enables the non-invasive analysis of oxygen metabolism and tissue hypoxia as well as associated neovascularization and microvascular architecture. However, evaluating such complex neuroimaging data requires computational support. Traditional machine learning algorithms and simple deep learning models were trained with radiomic features from clinical MRI (cMRI) or physio-metabolic MRI data. A total of 215 patients (first center: 166 participants + 16 participants for independent internal testing of the algorithms versus second site: 33 participants for independent external testing) were enrolled using two different physio-metabolic MRI protocols. The algorithms trained with physio-metabolic data demonstrated the best classification performance in independent internal testing: precision, 91.7%; accuracy, 87.5%; area under the receiver operating curve (AUROC), 0.979. In external testing, traditional machine learning models trained with cMRI data exhibited the best *IDH* classification results: precision, 84.9%; accuracy, 81.8%; and AUROC, 0.879. The poor performance for the physio-metabolic MRI approach appears to be explainable by site-dependent differences in data acquisition methodologies. The physio-metabolic MRI approach potentially supports reliable classification of *IDH* gene status in the presurgical stage of glioma patients. However, non-standardized protocols limit the level of evidence and underlie the need for a reproducible framework of data acquisition techniques.

## 1. Introduction

Glioma represents the most common primary brain tumor and arises from glial cells that surround and support neurons such as astrocytes, oligodendrocytes, and ependymal cells. Treatment planning for glioma depends on the World Health Organization (WHO) classification system of central nervous system (CNS) tumors, with gliomas designated as “low grade” (WHO grade 1 or 2) or “high grade” (WHO grade 3 or 4) based on the growth potential and aggressiveness of the tumor. Glioblastoma (WHO grade 4) is the most aggressive primary brain tumor in adults. The median glioblastoma survival time is 14–17 months [1]. Only 25% of patients survive more than one year after initial diagnosis and only 5% survive more than five years [2]. The latest (fifth) edition of the WHO classification system of CNS tumors published in 2021 [3] considered the isocitrate dehydrogenase (*IDH*) gene mutation status relevant for the diagnostic principles and nomenclature, with important implications for clinical practice such as treatment, patient prognosis, and the conceptualization/interpretation of clinical trials.

In the new classification, glioblastomas comprise only *IDH*-wildtype tumors. In addition, *IDH*-wildtype diffuse astrocytic tumors in adults without the histologic features of glioblastoma but with one or more of three genetic parameters (TERT promoter mutation, EGFR gene amplification, or combined gain of entire chromosome 7 and loss of entire chromosome 10) are also classified as glioblastoma. This has important implications for the prognosis and therapy of *IDH*-wildtype astrocytoma WHO grade 2 or 3. On the other hand, *IDH*-mutant gliomas are characterized by a class-defining and possibly tumor-initiating clonal *IDH*1 or *IDH*2 gene mutation [4,5]. These encompass astrocytomas (*IDH*-mutant, WHO grade 2, 3, or 4) and oligodendrogliomas (*IDH*-mutant and 1p/19q-codeleted, WHO grade 2 or 3) [3]. Patients with *IDH*-mutant glioma have better clinical outcomes compared with patients suffering from *IDH*-wildtype glioma [5,6], suggesting that there are unique physiological characteristics of *IDH*-mutant gliomas that may make them particularly vulnerable to specific therapies. The *IDH* gene mutation arises early during gliomagenesis, persists throughout the lifespan of the tumor, and confers neomorphic enzymatic activity [7], promoting cellular metabolism changes [8,9,10]. Like other malignant tumors, *IDH*-wildtype glioma is characterized by a high level of aerobic glycolysis, also known as the Warburg effect. Hypoxia induces upregulation of the transcription factor HIF1α, leading to overexpression of glycolysis-related proteins and enzymes, thereby accelerating glycolysis. In *IDH*-mutated glioma, however, the HIF1α response to hypoxia is blunted, shifting the metabolism to oxidative phosphorylation. A ^13^C metabolic flux analysis [10] on a panel of *IDH*-mutated glioma cell lines observed increased oxidative metabolism in the Krebs cycle as well as reduced tumor growth rates under hypoxic conditions, suggesting *IDH*1 mutant gliomas prefer a more oxygenated microenvironment for proliferation. This was confirmed by the results of an imaging study in glioma patients [11] using simultaneous pH- and oxygen-sensitive molecular magnetic resonance imaging (MRI), which suggest that mutation of the *IDH* gene is associated with lower tumor acidity and lower vascular hypoxia.

This separation into *IDH*-wildtype and *IDH*-mutated gliomas is an important advance. However, it places particular emphasis on neuropathology laboratories having access to adequate molecular testing and the ability to obtain results in a timely manner.

Molecular imaging is a promising method for non-invasively investigating the metabolic features of different types of tumors, permitting the distinction between tumor progression and treatment effects [12]. The known physiological connection between neovascularization and tissue hypoxia in brain tumors [13] reflected our rationale for developing a physio-metabolic MRI mapping tool for a combined quantitative assay of neovascularization activity, microvascular architecture, oxygen metabolism, and CNS tissue hypoxia [14,15,16].

In order to obtain deeper insight into the pathophysiology of brain tumors, our approach combines multiparametric quantitative blood oxygenation level-dependent (qBOLD) MRI [17] and vascular architecture mapping (VAM) [18]. The physical basis for the VAM approach is the vessel diameter-dependent sensitivity of spin echo and gradient echo MRI sequences [19]: spin echo perfusion measurements exhibit a peak sensitivity to the microvasculature at a vessel diameter of approximately 10 mm, whereas gradient echo perfusion measurements, which are commonly used in clinical routine, are dominated by larger vessel diameters (>20 mm) [19]. Consequently, the skillful combination of both spin echo and gradient echo perfusion MRI, as used in our physio-metabolic MRI approach, allows for the examination of the entire physiological and pathophysiological range of vessel diameters and structures in the human brain.

Inclusion of innovative neuroimaging methods into clinical routine, however, requires the assessment of a large amount of additional complex neuroimaging data and is time consuming. A timely evaluation of this helpful diagnostic information requires computational support. Methods from artificial intelligence (AI) such as deep learning (DL) and traditional machine learning (ML) may offer new options for timely support of clinicians. AI-based analysis is independent of the individual experience level of the evaluating clinician, helps to increase the comparability of results, and makes it possible to cope with large amounts of data. Finally, radiomics extracts semi-quantitative and quantitative features from neuroimaging data that are beyond human perception [20,21,22].

In recent years, deep learning has developed tremendously and has become widely used in various fields such as natural language processing or image processing [23,24]. Convolutional neural networks (CNNs) execute tasks by scanning the data with various filters in layers for convolution, pooling, and normalization to extract abundant features [25]. CNNs, however, are limited in learning dependencies between distant data points; thus, it is challenging for CNNs to directly learn the most important and representative features from data. Recurrent neural networks (RNNs) are deep learning models typically used to solve problems with longer sequential input data such as time series or radiomic feature vectors [26,27], thereby circumventing the mentioned limitations of CNNs. However, one of the biggest challenges with RNNs is vanishing or exploding gradients. These occur when the gradients of the loss function with respect to the parameters of the RNN become very small or very large as they propagate through the sequential input data. This can make it difficult to train the RNN effectively, as the updates to the parameters will be very small or very large and the network will not learn effectively. Long short-term memory (LSTM) is an RNN architecture [28,29] that has been widely adopted for sequential data processing as its design specifically solves the gradient vanishing/exploding problem of RNNs and aims to provide a short-term memory for RNNs that can last thousands of time steps, hence the name “long short-term memory”. LSTMs have been employed for medical image analysis recently [30,31]. For example, Zou et al. [32] used an LSTM model for pharmacokinetic parameter estimation in head and neck cancer using a dynamic contrast-enhanced MRI sequence.

Building on these findings and our preliminary work, we aimed to determine whether the combined use of deep learning algorithms and physio-metabolic MRI data enables a reliable preoperative classification of the *IDH* gene mutation status of glioma patients in a clinical setting.

To test this hypothesis, traditional ML algorithms and deep learning algorithms including a CNN and an LSTM were trained with conventional MRI data and physio-metabolic MRI data for oxygen metabolism and neovascularization from 166 consecutive glioma patients. The algorithms were tested with data from an independent internal test cohort as well as with data from an independent external test cohort using a different MRI data acquisition protocols.

## 2. Materials and Methods

### 2.1. Patients

The prospectively populated institutional MR neuroimaging databases of two sites were searched to identify patients with untreated brain tumors that were newly diagnosed between January 2016 and May 2023 (University Clinic St. Pölten) and June 2016 and October 2018 (University Clinic Erlangen), respectively. Inclusion criteria were as follows: (i) age ≥ 18 years; (ii) histopathological confirmation of adult-type diffuse glioma based on the 2021 WHO Classification of Tumors of the CNS [3], including conclusive information about isocitrate dehydrogenase 1 and 2 (*IDH*1 and 2) gene mutation status, i.e., glioblastoma *IDH*-wildtype WHO grade 4, astrocytoma *IDH*-mutant WHO grade 2–4, or oligodendroglioma *IDH*-mutant WHO grade 2–3; (iii) no previous treatment of the brain tumor; and (iv) a preoperative MRI examination with the study protocol.

The brain tumor MRI database at the University Clinic St. Pölten contained a total of 2156 MRI examinations according to the study protocol in 681 brain tumor patients for the period from January 2016 to May 2023. A total of 182 patients (89 females; 93 males; mean age 57.5 ± 15.5 years; 19–89 years) with newly diagnosed, untreated glioma fulfilled/satisfied the inclusion criteria.

Patients examined from January 2016 to January 2023 at the University Clinic St. Pölten were selected as the training cohort. Of the total of 166 patients (81 females; 85 males; mean age 58.1 ± 15.2 years; 19–89 years) during this period, 123 patients (74%; 59 females; 64 males; mean age 63.2 ± 12.3 years; 23–89 years) were diagnosed with a glioma without mutation of the *IDH* gene (*IDH*wt), and 43 patients (26%; 22 females; 21 males; mean age 43.3 ± 12.8 years; 19–73 years) were diagnosed with a glioma with mutation of the *IDH* gene (*IDH*mut). The difference in mean age between these two groups was significantly different (*p* < 0.001), as determined by a Student’s *t* test.

Of the 123 patients with a diagnosis of an *IDH*wt glioma, 116 patients had a glioblastoma WHO grade 4, five patients had an astrocytoma WHO grade 3, and two patients had an astrocytoma WHO grade 2. The 43 patients suffering from *IDH*-mutated glioma had the following histological diagnoses: eleven patients had an oligodendroglioma WHO grade 2, five patients had an oligodendroglioma WHO grade 3, six patients had an astrocytoma WHO grade 2, thirteen patients had an astrocytoma WHO grade 3, and eight patients had an astrocytoma WHO grade 4.

Patients who were examined at the University Clinic St. Pölten from February 2023 to May 2023 were selected as an independent internal test cohort. Of the total 16 patients (eight females; eight males; mean age 51.9 ± 18.6 years; 19–76 years) during this time period, twelve patients (75%; five females; seven males; mean age 53.4 ± 20.8 years; 19–76 years) had a diagnosis of *IDH*wt glioma (ten glioblastoma, one astrocytoma WHO grade 3, and one astrocytoma WHO grade 2) and four patients (25%; three females; one male; mean age 47.3 ± 10.9 years; 31–54 years) had a diagnosis of *IDH*mut glioma (two oligodendroglioma WHO grade 2 and two oligodendroglioma WHO grade 3).

All patients examined at the FAU Erlangen–Nürnberg (from June 2016 to October 2018) who met the inclusion criteria were selected as an independent external test cohort. Of the total 33 patients (seven females; twenty-six males; mean age 54.7 ± 15.3 years; 27–78 years), 21 patients (64%; four females; seventeen males; mean age 62.7 ± 10.5 years; 45–78 years) had a diagnosis of *IDH*wt glioma (twenty glioblastoma and one astrocytoma WHO grade 2) and twelve patients (36%; three females; nine males; mean age 40.6 ± 11.6 years; 27–67 years) had a diagnosis of *IDH*mut glioma (four oligodendroglioma WHO grade 3, three astrocytoma WHO grade 2, three astrocytoma WHO grade 3, and two astrocytoma WHO grade 4). The characteristics of the patient cohorts are summarized in Appendix A.

### 2.2. MRI Data Acquisition

#### 2.2.1. The MRI Study Protocol at the University Clinic St. Pölten

All data of the MRI study protocol were acquired on a clinical 3 Tesla scanner (Trio, Siemens, Erlangen, Germany) using the conventional 12-channel head coil provided by the vendor.

Clinical MRI (cMRI) for routine diagnosis of brain tumors included, among others, an axial fluid-attenuated inversion–recovery (FLAIR) sequence, high-resolution pre- and post-contrast enhanced T1-weighted (ceT1w) sequences, an axial diffusion-weighted imaging (DWI) sequence, and a gradient echo dynamic susceptibility contrast (GE-DSC) perfusion MRI sequence with 60 dynamic imaging volumes during administration of 0.1 mmol/kg-bodyweight gadoterate–meglumine (Dotarem, Guerbet, Villepinte, France).

MRI-based oxygen metabolism was assessed using the quantitative blood-oxygen-level-dependent (qBOLD) imaging approach [16], which additionally included the acquisition a multi-echo GE sequence and a multi-echo SE sequence to map the transverse relaxation rates, R_2_* (=1/T_2_*) and R_2_ (=1/T_2_), respectively.

MRI-based assessment of microvascular architecture and neovascularization activity was performed with the vascular architecture mapping (VAM) technique [18], which additionally included a spin echo DSC (SE-DSC) perfusion MRI sequence with the same parameters and contrast agent injection protocol as described for the routine GE-DSC perfusion MRI. Adverse effects due to contrast media leakage and differences in time to first-pass peak that may significantly affect the data evaluation were minimized as described previously [33,34].

All MRI sequences for qBOLD and VAM were performed with identical geometric parameters (number of slices, voxel size, etc.) and slice position as the routine GE-DSC perfusion sequence. The sequences required a total of seven minutes of additional scanning time.

#### 2.2.2. The MRI Study Protocol at the FAU Erlangen-Nürnberg

The MRI study protocol at the FAU Erlangen-Nürnberg differed from that at the University Clinic St. Pölten. The sequence parameters at the two sites are compared in Appendix A. MRI data acquisition at the FAU was also performed with a 3 Tesla scanner (Trio, Siemens, Erlangen, Germany), but in combination with a 32-channel head coil.

The cMRI protocol for routine diagnosis at the FAU also included axial FLAIR, ceT1w, and DWI sequences. Instead of performing the GE-DSC and the SE-DSC sequences separately with two administrations of contrast media, a hybrid single-shot gradient echo–spin echo (GESE) DSC sequence was used at the FAU. GESE-DSC perfusion examinations were performed with 80 dynamic measurements during administration of a double dose (0.2 mmol/kg bodyweight) of gadoterate meglumine (Dotarem, Guerbet, France). The GESE-DSC raw data were then separated into a GE-DSC and an SE-DSC dataset, respectively. We refer to Appendix A for more details.

This GESE-DSC sequence has the advantage that two separate SE- and GE-DSC acquisitions are not required. This reduces sensitivity to patient motions and variations in injection timing. The separate acquisition of the SE and GE-DSC sequences, on the other hand, enables the measurement of VAM data with a high signal-to-noise ratio (SNR), high spatial resolution, and coverage of the entire brain. This is due to the fact that with separate acquisition, the full data acquisition potential is available exclusively for the respective sequence and does not have to be shared. The combined GESE-DSC sequence is therefore less suitable for use in clinical routine.

The sequences for the qBOLD imaging approach for MRI-based oxygen metabolism assessment differed at the FAU, especially in terms of repetition times (TR) and the number of slices (Appendix A). TR has an especially significant impact on the data.

### 2.3. MRI Data Processing and Calculation of MRI Biomarker Maps

At both sites, MRI data processing and calculation of MRI biomarker maps were performed using in-house software (PrecisionMRI version 6.1 beta) developed with MatLab (MathWorks, Natick, MA, USA). Processing of the cMRI data included calculation of cerebral blood flow (CBF) and volume (CBV) maps from the GE-DSC perfusion MRI data using automatic identification of the arterial input functions (AIFs) [35,36]. Furthermore, apparent diffusion coefficient (ADC) maps were calculated from DWI data using a mono-exponential model (two-point b values of 0 and 1000 s/mm^2^) and the following equation:(1)$$\mathrm{ADC}=-\frac{1}{b}\;\cdot \;\ln\left(\frac{S}{S_{0}}\right)$$
where S_0_ is the signal intensity of no diffusion gradients (b = 0 s/mm^2^), S is the signal intensity with diffusion gradients (b = 1000 s/mm^2^), and b is the b value.

Processing of qBOLD data for MRI-based oxygen metabolism included corrections for background field artifacts in the R_2_*-mapping data [37] and for stimulated echoes in the R_2_-mapping data [38], followed by calculation of R_2_* and R_2_ maps. Biomarker maps of tissue oxygen metabolism including the oxygen extraction fraction (OEF) and cerebral metabolic rate of oxygen (CMRO_2_) [39] were calculated using the following equations:(2)$$\mathrm{OEF}=\frac{R_{2}^{*}-R_{2}}{\frac{4}{3}\cdot \pi \cdot \gamma \cdot \Delta \chi \cdot \mathrm{Hct}\cdot B_{0}\cdot \mathrm{CBV}}$$
and
(3)$${\mathrm{CMRO}}_{2}=\frac{C_{a}\;\cdot \;\mathrm{CBF}}{k\;\cdot \;\mathrm{CBV}}\cdot \left(R_{2}^{*}-R_{2}\right)$$
where Δχ = 0.264·10^−6^ is the difference between the magnetic susceptibilities of fully oxygenated and fully deoxygenated haemoglobin; Hct = 0.42·0.85 is the microvascular hematocrit fraction (the factor 0.85 stands for a correction factor of systemic Hct for small vessels); γ = 2.67502·10^8^ rad/s/T is the nuclear gyromagnetic ratio; and Ca = 8.68 mmol/mL is the arterial blood oxygen content [40]. Maps of capillary oxygen tension (capiPO_2_) and mitochondrial oxygen tension (mitoPO_2_) [41,42] were determined with:(4)$${c\mathrm{ap}\mathrm{iPO}}_{2}={\;P}_{50}\sqrt[{h}]{\left(\frac{2}{\mathrm{OEF}}-1\right)}$$
and
(5)$${\mathrm{mitoPO}}_{2}={\;P}_{50}\sqrt[{h}]{\left(\frac{2}{\mathrm{OEF}}-1\right)}-\frac{{\mathrm{CMRO}}_{2}}{L}$$
where h is the Hill coefficient of oxygen binding to hemoglobin (h = 2.7), P_50_ is the hemoglobin half-saturation tension of oxygen (27 mmHg), and L (4.4 mmol/Hg per minute) is the tissue oxygen conductivity as defined by Vafaee and Gjedde [43].

Processing VAM data for MRI-based microvascular architecture and neovascularization activity included correction for remaining contrast agent extravasation as described previously [33,34], fitting of the first bolus curves for each voxel of the GE- and SE-DSC perfusion MRI data with a previously described gamma-variate function [44], and calculation of the the ∆R_2,GE_ versus (∆R_2,SE_)^3/2^ diagram, also known as the vascular hysteresis loop (VHL). [45]. Biomarker maps of microvascular architecture [46] including the microvessel density (MVD) and vessel size index (VSI, i.e., microvessel radius) were calculated using the VHL curve data and the following equations:(6)$$\mathrm{MVD}=\frac{Q_{\max}}{\beta }\cdot {\;\left(\frac{{\mathrm{CBV}}^{2}}{4\pi^{2}\cdot \mathrm{ADC}\cdot {\overset{-}{R}}^{4}}\right)}^{1/3}$$
and
(7)$$\mathrm{VSI}={\left(\frac{\mathrm{CBV}\cdot \mathrm{ADC}\cdot {\;\beta }^{3}}{2\pi \cdot Q_{\max}^{3}}\right)}^{1/2}$$
where Q_max_ = max[∆R_2,GE_]/max[(∆R_2,GE_)^3/2^], β is a numerical constant (β = 1.6781), and ${\overset{-}{R}}^{4}$ ≈ 3.0 μm is the mean vessel lumen radius [46]. Neovascularization activity was estimated by the microvessel type indicator (MTI), which was previously [18] defined as the area of the VHL curve signed with the rotational direction of the VHL curve (a clockwise VHL direction was identified with a plus-sign and a counter-clockwise VHL direction was identified with a minus-sign [18]). Previous studies [18,47] demonstrated that a positive MTI value (assigned to warm colors in the MTI maps) is associated with a vascular system that is dominated by arterioles, whereas a negative MTI value (cool colors in the MTI maps) is associated with venule- and capillary-like vessel components. Finally, the map for the microvascular cerebral blood volume (μCBV) was calculated from the SE-DSC perfusion MRI data via a separate automatic identification of AIFs [36].

In summary, MRI data processing resulted in the following data sets for each patient for further data analysis:Four cMRI data sets: FLAIR and ceT1w MRI data as well as the ADC maps for microstructural density and the CBV maps for macrovascular perfusion.Four biomarker maps for oxygen metabolism: MRI-based tissue oxygen metabolism (OEF and CMRO_2_) as well as MRI-based capillary oxygen tension (capiPO_2_) and mitochondrial (tissue) oxygen tension (mitoPO_2_).Four biomarker maps for microvascular architecture and neovascularization activity: microvascular density (MVD), microvascular diameter (VSI), microvascular perfusion (µCBV), and microvascular type (MTI).

### 2.4. Radiomic Feature Extraction

The data processing pipeline is summarized in Appendix A, and the procedures have been described in more detail in the Appendix A as well as in previous publications [48]. Summarized briefly, the anatomical cMRI data and biomarker maps of a patient were loaded into the open-source software platform 3D Slicer (v. 4.11). The tumor volume was segmented on ceT1w and FLAIR MRI data, respectively. The anatomical MRI data were z-score normalized [49,50], resampled into a uniform voxel size of 1 × 1 × 1 mm^3^, and grey-level discretization was performed [51] with a bin width value of 0.1 resulting in histograms with approximately 60 bins. Biomarker maps, however, represented quantitative imaging data with a range of physiological reasonable values. Individually adapted thresholds were applied to the biomarker maps, and biomarker value discretization was performed with adapted bin width values in order to obtain histograms with 60–67 bins. Appendix A summarizes the value ranges and bin sizes used for the biomarker maps.

Next, 107 radiomic features were extracted for every data set from the segmented tumor volume using the Python package PyRadiomics [52], with 14 shape features, 18 first-order features, and 75 texture features, respectively. A detailed overview and description of these radiomic features can be found in the Appendix A and elsewhere [48]. Procedures and features were in accordance with the Imaging Biomarker Standardization Initiative (IBSI) [53]. Mathematical formulas have been described on the website of the package (https://pyradiomics.readthedocs.io; accessed on 22 February 2023). The feature vectors for ceT1w, FLAIR, ADC, and CBV were concatenated to the cMRI feature vector, those for OEF, CMRO_2_, capiPO_2_, and mitoPO_2_ were concatenated to the feature vector of oxygen metabolism, and the feature vectors for vasculature and neovascularization included the features for µCBV, MVD, VSI, or MTI. Thus, all three of these resulting feature vectors contained 428 features per patient.

### 2.5. Traditional Machine Learning

Traditional machine learning, including radiomic feature selection and model development, was performed using the open-source software package Weka (version 3.8.5, University of Waikato, Hamilton, New Zealand). For the selection of radiomic features, we proceeded in two steps and followed the strategy combining advantages of both the ranking methods and the learner-based methods as described previously [54].

Firstly, we applied the RefiefF attribute evaluation filter in combination with the Ranker search method using the following parameters: number of nearest neighbors for attribute estimation = 10; random seed for sampling instances; sigma that sets the influence of nearest neighbors = 2; and no weighting of nearest neighbors by their distance. The top 25% of features in the ranking were selected and used in the next step.

Secondly, we applied the learner-based feature selection method Wrapper using the respective ML algorithm that was used for model development (we refer to the next chapter for a detailed description) as classifier. The BestFirst search method was used in combination with 10-fold cross-validation. Furthermore, we used accuracy as a measure to evaluate the performance of attribute combinations and a threshold of 0.01. Those features that were selected at least once in the rankings were selected as the most suitable features specifically for each ML algorithm, i.e., an individual feature list was generated for each ML classifier.

As we expected unbalanced classes due to different prevalence and well-known differences in patient numbers at our institution for the brain tumor entities, the synthetic minority oversampling technique (SMOTE) [55] was employed in order to balance the classes using five nearest neighbors and one seed for random sampling. This was followed by the application of a randomize filter.

For model development, only those algorithms were considered that had shown top performance in brain tumor classification in a previous study, namely:A multilayer perceptron (MLP) with one hidden layer and number of neurons = number of features + number of classes;Adaptive boosting (ABoost) using decision tree “J48” as classifier; andRandom forest (RF).

Short descriptions of these traditional ML algorithms are summarized in the Appendix A. A list of optimized hyperparameters is provided in Appendix A. Four categories of data were used as training data sets:cMRI (ceT1w, FLAIR, ADC, CBV);MRI-based oxygen metabolism (CMRO_2_, OEF, capiPO_2_, mitoPO_2_);MRI-based vascular architecture (µCBV, MVD, MTI, VSI); andThe combination of MRI-based oxygen metabolism and vascular architecture.

A 10-fold cross-validation procedure was used to train and validate the models.

### 2.6. Deep Learning

For the development of the deep learning algorithms, i.e., 1D-CNN and LSTM networks, the open-source data analysis platform KNIME (Konstanz Information Miner; version 4.7.4) was used, which is widely known and performs well in biomedical and healthcare applications [56,57,58]. The input data for the deep learning algorithms were the full radiomic feature vectors for cMRI, MRI-based oxygen metabolism, and MRI-based vascular architecture, respectively. Each feature vector included 428 (4 × 107) features. We additionally used the combination of MRI-based oxygen metabolism and vascular architecture, which included 856 (2 × 428) features.

CNNs are the most widely used architectures in deep learning approaches and are generally composed of an input layer, several hidden layers for convolution, pooling, and fully connected “dense” layers, and an output layer. Our CNN for prediction of the *IDH* gene mutation status of glioma was based on a one-dimensional CNN (1D-CNN) [59]. It was characterized by convolution kernels that slide in only one dimension over the elements of the input pattern, i.e., the feature vectors for cMRI, oxygen metabolism, vascular architecture, or the combination of the latter two. The architecture of the 1D-CNN included nine hidden layers: three convolutional layers, three max-pooling layers, one flatten layer, and two fully connected (dense) layers. A rectified linear unit (ReLU) was used as an activation function for all convolutional layers and the first dense layer [60]. A sigmoid function was applied in the final dense layer for activation and calculation of an output with one unit for binary classification: *IDH*-wildtype (*IDH*wt) or *IDH*-mutant (*IDH*mut). After the second and third convolution layer as well as after the first dense layer, three dropout rates of 0.2 were used in order to avoid overfitting [61]. The binary cross-entropy was considered a loss function, and Adam, an adaptive learning rate optimization algorithm specifically designed for deep neural network training, was exploited as an optimizer. The parameters are summarized in Appendix A. The CNN was trained with 80% and validated with 20% of the SMOTE balanced training data (the same as used for the ML models). The data were shuffled before each epoch (up to 300) in order to ensure the robustness of the models.

LSTMs learn (long) distant dependencies within the sequential input data using a gating mechanism that controls the memorizing process. An LSTM unit is composed of a cell that can store, read, and write information via a forget gate, an input gate, and an output gate that regulate the flow of information into and out of the cell. The forget gate decides which information from a previous state needs attention and which can be ignored. The input gate learns new sets of information from the incoming cell and decides which pieces of new information to store in the current state. The output gate controls which pieces of information in the current state to output as the next hidden state.

The inputs to LSTM were again the combined feature vectors from the cMRI data, MRI-based oxygen metabolism, and MRI-based vascular architecture, respectively, of the *IDH*wt and *IDH*mut glioma, with 4 × 107 (428) features each.

The LSTM models consisted of an input layer, four LSTM [62] layers (eight layers for the combination of MRI-based oxygen metabolism and vascular architecture), and two fully connected (dense) layers. The serial direction of an LSTM layer was occupied by the 107 radiomic features from one of the biomarker volumes. For all LSTM layers, the hyperbolic tangent (tanh) was the activation function, the hard sigmoid was the recurrent activation function, and 0.2 was the forward and recurrent dropout rate, respectively. The dense layers applied a sigmoid function for activation with a dropout rate of 0.2. The second dense layer had an output with one unit for binary classification of *IDH*wt or *IDH*mut. The LSTM network was also trained with 80% and validated with 20% of the SMOTE balanced training data using the Adam optimizer with an initial learning rate of 0.001, beta1 of 0.9, beta2 of 0.999, and epsilon of 10^−8^. The training batch size was set to 28, the validation batch size was set to 10, and each adapted to the number of cases in the training and validation data sets. The number of epochs was 300. A list of optimized hyperparameters is provided in Appendix A.

### 2.7. Model Performance Testing

Data from the University Clinic St. Pölten were used as independent internal test cohort, and data from the FAU Erlangen-Nürnberg were used as independent external test cohort with different MRI examination protocols. For performance evaluation, confusion matrix-derived metrics including accuracy, sensitivity (aka the recall or true positive rate), specificity (aka the true negative rate), precision (aka the positive predictive value), and the F-score, as well as the area under the receiver operating characteristic curve (AUROC) [63], were calculated. Details are provided in Appendix A. Because the data sets for both independent internal and independent external testing were unbalanced, we calculated the weighted mean over the performance metrics for the *IDH*-wildtype and *IDH*-mutated gliomas.

## 3. Results

### 3.1. The Selected Radiomic Features

The feature selection in the training data for the traditional ML algorithms resulted in the following feature distributions. For cMRI: 62% ceT1w features, 1% FLAIR features, 16% ADC features, and 21% CBV features. For MRI-based oxygen metabolism: 32% CMRO2 features, 9% OEF features, 27% capiPO2 features, and 32% mitoPO2 features. For MRI-based vascular architecture: 14% µCBV features, 37% MVD features, 31% MTI features, and 18% VSI features.

During training, the loss curves of all selected algorithms showed that the training models converged. The accuracy values for the validation of the trained models are summarized in Table 1. The LSTM algorithm in combination with the cMRI data showed the highest accuracy in validating the trained models, followed by the MLP with the biomarkers for oxygen metabolism and the RF model with the data for vascular architecture, respectively. In general, MLP and RF showed the best accuracy in training validation, while ABoost and the CNN showed the worst accuracy.

### 3.2. Testing with an Independent Internal Cohort

For the independent internal test cohort of 16 patients examined at the University Clinic St. Pölten, both the LSTM trained with the MRI-based oxygen metabolism and the MLP trained with the combination of oxygen metabolism and vascular architecture showed the best classification performance: two errors (2 *IDH*wt glioma), an AUROC of 0.979, an accuracy of 0.875, and a precision of 0.917. The CNN trained with the data for vascular architecture and the RF model trained with the combination of oxygen metabolism and vascular architecture also showed only two errors, although one was for *IDH*wt glioma and one was for *IDH*mut glioma. This means that all of the best-performing algorithms were trained with physio-metabolic MRI data. Furthermore, most algorithms (five out of six) with medium classification quality (three errors) were also trained with physio-metabolic data. Only the LSTM trained on cMRI data also showed medium-rank classification quality. Poor classification quality (four and five errors out of 16 patients) in the tests with the independent internal cohort was particularly evident in algorithms trained with cMRI data (four out of five). However, six out of fifteen algorithms trained with physio-metabolic data also showed poor classification quality when tested with the independent internal cohort. These primarily included CNN in combination with oxygen metabolism and ABoost in combination with VAM.

The values for accuracy, sensitivity, specificity, precision, F-score, AUROC, and classification errors for the five *IDH* gene status classifiers and the four MRI data sets (cMRI; oxygen metabolism; VAM; combination of oxygen metabolism & VAM) when tested with the independent internal test cohort are presented as heat maps in Figure 1. Confusion matrices are provided in Appendix A.

Representative cases of the independent internal patient testing cohort for patients with an *IDH*-mutated glioma and an *IDH*-wildtype glioma that were correctly classified by all algorithms are shown in Figure 2 and Figure 3, respectively. Figure 4, on the other hand, shows the case of a patient who was misclassified by all algorithms due to strong motion artifacts in the perfusion and relaxometry MRI sequences.

### 3.3. Testing with an Independent External Cohort

The tests of the algorithms with the independent external patient cohort from the FAU Erlangen-Nürnberg produced completely opposite results compared to the tests with the independent internal cohort. Algorithms trained with the cMRI data from the University Clinic St. Pölten showed the best classification quality when tested with the independent external cMRI data from the FAU Erlangen-Nürnberg. ABoost performed best with six errors out of 33 patients, a precision of 0.849, and an accuracy of 0.818. All other algorithms trained with cMRI data followed with seven to nine errors in the middle range. Only the MLP algorithm, which was trained with the oxygen metabolism and VAM data combination, and the ABoost model, which was trained with oxygen metabolism, were also in the middle range of classification quality, with seven and nine errors. All other algorithms that were trained with physio-metabolic data from St. Pölten showed poor classification quality, with 10 to 13 errors in the 33 patients when tested with the independent external data from FAU Erlangen-Nürnberg.

The values for the quality parameters of the five *IDH* gene status classifiers and the four MRI data sets when tested with the independent internal test cohort are presented as heat maps in Figure 5. Confusion matrices are provided in Appendix A.

Cases of the independent external patient cohort that were misclassified by all algorithms are depicted in Figure 6 and Figure 7, respectively. The reasons for these misclassifications were MRI signal changes that were atypical for an *IDH*-wildtype glioblastoma (Figure 6) and susceptibility artifacts due to bleeding in the *IDH*-mutated oligodendroglioma WHO grade 3 (Figure 7), respectively.

## 4. Discussion

Early and reliable characterization of the *IDH* gene mutation status of glioma is crucial for personalized treatment decisions and prognosis in clinical neurooncology. The gold standard, histopathological analysis of invasively obtained tissue samples, requires additional time and financial resources. A reliable preoperative, non-invasive determination of *IDH* gene status would therefore be of great benefit.

In this study, we trained traditional ML algorithms and simple DL models with radiomic features from both physio-metabolic MRI data and routine MRI data based on differences in energy metabolism between *IDH*mut and *IDH*wt gliomas. The algorithms trained with physio-metabolic data showed the best classification performance when tested with an independent internal cohort, with precision, accuracy, and AUROC up to 91.7%, 87.5%, and 0.979, respectively. The LSTM model in combination with oxygen metabolism and the MLP in combination with oxygen metabolism along with microvascular architecture demonstrated the highest performance parameters, which in part was in line with our hypothesis: combining physio-metabolic MRI and deep learning algorithms enables a noninvasive and reliable classification of the *IDH* gene mutation status of gliomas.

In contrast, CNN showed poor classification performance for *IDH* status in combination with oxygen metabolism, which can be explained by the well-known limitations of CNNs in learning dependencies between distant data points, which appear to be particularly relevant to the radiomic features of the biomarkers for oxygen metabolism, i.e., OEF, CMRO_2_, capiPO_2_, and mitoPO_2_. However, when combined with the microvascular architecture features, CNN performed very well, indicating greater redundancy of the biomarkers (µCBV, MVD, MTI, VSI). Training the DL algorithms (CNN and LSTM) with the combination of oxygen metabolism plus microvascular architecture resulted in a significant deterioration/reduction in diagnostic performance, which in turn may be caused by the extended feature vector (856 features) for the relatively simple DL models, while MLP and RF performed very well when trained on the combined OxyMet plus VAM data.

To our surprise, the results for independent external testing were clear, with the models trained with the cMRI data from UK St. Pölten consistently yielding the best results for *IDH* classification for the external test cohort from FAU Erlangen with precision, accuracy, and AUROC of up to 84.9%, 81.8%, and 0.879. The poor performance of the models (ML and DL) trained with the physio-metabolic data are related to relevant differences in the innovative physiologic and metabolic MRI techniques for data acquisition between the study sites.

These results also reflect a limitation in applying AI models in combination with experimental imaging data for classification. There is often no uniform standard for experimental imaging methods. Rather, each institution often follows its own technical approach, sometimes with major differences in MR technology, protocols, and post-processing. This means a great challenge for multicenter studies and building up large public databases.

A growing number of studies have been published using traditional ML or DL methods combined with imaging data in order to classify glioma *IDH* gene status and can essentially be divided into two groups: studies that used innovative imaging techniques in combination with traditional ML or simple DL algorithms and studies that used exclusively anatomic MRI data from large, publicly available, multicenter databases in combination with advanced DL models. Our study clearly belongs to the former group. The previously published papers in this group, however, have not independently tested their algorithms on data unseen by the model.

There exists a broad variety of experimental neuroimaging methods, with only two studies using metabolic imaging methods for *IDH* classification.

Ozturk-Isik et al. [64] used short-echo time proton MR spectroscopy (1H-MRS) to examine metabolite levels in 112 glioma patients. They used this metabolic information in combination with traditional ML algorithms for noninvasive preoperative classification of *IDH* mutation subgroups. Using the logistic regression algorithm, they were able to predict the presence of an *IDH* mutation with an accuracy of 88.4%. Tatekawa et al. [65] developed a method for voxel-wise clustering of multiparametric MRI and 18F-dihydroxyphenylalanine (FDOPA) positron emission tomography (PET) images using an unsupervised 16-class clustering approach. Combined with a support vector machine algorithm, they classified the *IDH*-gene mutation status of 62 glioma patients with a classification accuracy of 76% or an AUROC of 0.81, respectively. However, due to the small number of patients in these two studies, the authors did not carry out tests with an independent cohort, but only used leave-one-out cross-validation.

Kesler et al. [66] followed an interesting approach by determining gray matter connectomes from preoperative non-contrast T1w MRI scans and demonstrated that gray matter connectomes are useful for distinguishing between *IDH* variants of high-grade glioma. In a subsequent study [67], the authors extracted 93 connectome features from the preoperative T1w MRIs of 234 glioma patients and evaluated the performance of four traditional ML algorithms for predicting the *IDH* genotype. Using features for the connectome efficiency, brain volume, network degree, network size, tumor hemisphere, and tumor lobe in combination with an RF model, the authors achieved an accuracy and AUROC of 89% and 0.95, respectively. Based on the assumption that *IDH*-mutant gliomas have lower cellular density and increased interstitial edema, Karami et al. [68] evaluated the potential of multi-shell diffusion MRI in combination with transfer learning of a pretrained DL model for *IDH* gene status classification. The multi-shell diffusion MRI approach additionally uses significantly higher b-values, which enables more informative diffusion models. By adding these experimental data to cMRI data, the authors achieved an increase in accuracy from 74% to 81% for prediction of the *IDH* gene status in 146 glioma patients. However, in these two studies, no tests were carried out on independent cohorts, but rather leave-one-out [67] and five-fold cross-validation [68] were used, respectively. In our study, accuracy values for training validation ranged from 89.9% to 94.0%, which is significantly higher than for most previous studies using experimental imaging techniques. At this point, it is important to emphasize that independent testing of the algorithms may present an essential quality criterion for such studies, and we strongly recommend its use.

To our knowledge, the only study that has independently tested DL models using experimental MRI methods to classify *IDH* gene status is that of Choi et al. [69], who performed DSC perfusion MRI to obtain intensity-time curves of the T_2_*-susceptibility signal from tumor subregions segmented by a CNN. These T_2_*-time curves and the corresponding arterial input functions from 395 patients were used as multidimensional input to train a convolutional LSTM model with an attention mechanism. They were able to predict the *IDH* genotype in an independent test cohort of 50 patients with 91.7% accuracy. We also achieved this accuracy value with the best-performing models in our study, LSTM with oxygen metabolism and MLP with oxygen metabolism and VAM, respectively, but were only able to include half as many patients in our study. Data from more patients and the use of more up-to-date algorithms could further improve the performance of our approach. However, in the study by Choi et al. [69], non-independent external testing was carried out, which should be the standard for future studies on clinical AI applications.

Independent external testing has been carried out in most of the studies of the above-mentioned second group, i.e., studies that have performed the classification of the *IDH* gene status using anatomical MRI data from large publicly available databases in combination with more advanced DL methods. These studies usually used several of the following public databases: The Cancer Imaging Archive (TCIA) [70], the Repository of Molecular Brain Neoplasia Data (REMBRANDT) [71], the Clinical Proteomic Tumor Analysis Consortium Glioblastoma Multiforme (CPTAC-GBM) collection [72], the Ivy Glioblastoma Atlas Project (Ivy GAP) collection [73], the Brain Tumor Segmentation (BraTS) challenge dataset [74], and the Erasmus Glioma Database [75]. In addition, databases from the participating universities were often used for training and/or independent testing.

The three studies with the best performance in *IDH* classification used data from over 1000 patients in combination with advanced CNNs. Chakrabarty et al. [76] used a 2.5D hybrid CNN to simultaneously locate the glioma and classify its *IDH* status. By leveraging MRI features and knowledge features from clinical records and tumor location, the authors achieved an accuracy and AUROC of 93.5% and 0.925 for independent internal testing and 94.1% and 0.933 for independent external testing, respectively. Choi et al. [77] combined a 3D U-shaped CNN for tumor segmentation with a CNN classifier (34-layer ResNet) and achieved with this hybrid model an accuracy and AUROC of 93.8% and 0.96 for the internal test, as well as 87.9% and 0.94 for the external test. Finally, Bangalore Yogananda et al. [78] used a so-called nnU-Net architecture, which is characterized by excellent generalizability and reduced risk of overfitting by transferring information from each preceding layer to the subsequent ones. Additionally, the nnU-Net automatically configures itself, including pre-processing, network architecture, training, and post-processing [79]. In independent internal testing, they achieved an overall accuracy of 92.8% with an AUROC of 0.96. Other studies [80,81,82], which also used multi-site data from public databases but with fewer patients or which only used a simple CNN, achieved significantly lower accuracy values (84.0–85.7%).

The accuracy of the three studies [76,77,78] with the top performance in the independent internal testing (92.8–93.8%) was significantly higher than our best result (87.5%); the AUROC values (0.925–0.96), however, were lower than our result (0.979). Our independent internal testing cohort was quite small, at only 16 patients, which is a limitation of our study. However, during the prespecified four-month period for enrolling patients into the testing cohort, only these 16 patients with untreated gliomas were examined.

The results of the independent external tests clearly demonstrate the advantages of the approach of training more complex DL models with multi-site data from many patients from public databases. Chakrabaty et al. [76] and Choi et al. [77] achieved accuracies of 94.1% and 87.9% and AUROC values of 0.933 and 0.94, respectively, in their independent external tests. In our study, the traditional ML models trained with cMRI data achieved the best classification results, with an accuracy of max. 81.8% and AUROC of max. 0.879, respectively. The models trained with the physio-metabolic MRI data, especially the DL algorithms, performed far worse in our independent external testing. In addition to the insufficient generalization of the models due to training with single-site data, the differences in the data acquisition techniques of the physio-metabolic MRI data between the two sites led to a further increase in overfitting.

These case studies demonstrated that motion and susceptibility artifacts were also a reason for some misclassifications, as presented in our physio-metabolic MRI data. Susceptibility artifacts caused by intratumoral bleeding (Figure 7), for example, cannot be corrected. Nalawade et al. [83], however, demonstrated that classifier performance molecular glioma markers including *IDH* status decreased markedly with increasing motion corruption of the MRI data, but applying motion correction effectively restored classification accuracy for even the most motion-corrupted images.

There are some biasing concerns in our present study worthy of mention. Calculation of ADC values was performed by linear approximation with two-point b-values (b = 0 and 1000 s/mm^2^) and did not take into account higher b-values or the calculation of more quantitative biomarkers for diffusion. However, this procedure is still widely used in clinical routine diagnostics. As noted above, the patient cohorts for both training/validation (166 patients) and independent internal testing (16 patients) were quite small, particularly compared to studies using large-scale public databases. Additionally, all data were collected using a single MRI scanner at a single site, leading to insufficient generalization of the AI algorithms due to differences in MR scanner setups from different manufacturers, different magnetic field strengths, and variations in examination protocols that were not considered. The differences in data acquisition techniques for quantitative physio-metabolic MRI data between the two sites involved had a greater impact on classification performance than expected. The independent external testing clearly demonstrated the limitations of using AI models in combination with innovative imaging methods developed and applied at a single site. To prevent these pronounced effects of model overfitting, the construction of public databases for innovative imaging data from different sites with different settings is necessary.

Two recently published multicenter studies have shown how to prevent model overfitting. Prof. Marias’ group used DSC-MRI perfusion data from four MRI scanners at three different sites, from three different manufacturers, and operating at two different field strengths to predict the *IDH* gene mutation of gliomas. Data from one site were used for training and data from the other two sites were used for independent external testing. In the study by Manikis et al. [84], the authors were able to increase the accuracy of *IDH* status classification from 54.4% to 70.6% using dynamics-based standardization of images. In the study by Ioannidis et al. [85] using mathematical models to quantify the DSC data, the accuracy of *IDH* mutation prediction was increased to 75%.

Another possible solution to prevent the limited generalization of the applied models is to use self-supervised learning-based foundation models, which represent large AI models trained on a vast quantity of unlabeled data at scale, resulting in models that can be adapted to a wide range of downstream tasks [86]. The self-supervised learning approach reduces data inefficiency by deriving supervisory signals directly from data rather than relying on expert knowledge using labels [87,88]. Self-supervised learning uses huge amounts of unlabeled data to learn general-purpose feature representations that can be easily adapted easily to more specific tasks. After this pretraining phase, the models are fine-tuned to specific downstream tasks such as segmentation, classification, or prediction. Recently, Zhou et al. [89] presented a foundation model for retinal images using trained self-supervised learning on 1.6 million unlabeled retinal images. They adapted this foundation model to diagnosis and prognosis of sight-threatening eye diseases as well as incident prediction of complex systemic disorders such as heart failure and myocardial infarction, sometimes with just a few hundred labeled data. They demonstrated that the adapted foundation model provides a generalizable solution to improve model performance and consistently outperforms several traditional benchmark models. Creating a baseline model for brain imaging is time- and labor-intensive and requires powerful hardware resources and huge amounts of data. However, once generated and made publicly available, it could be adapted for a variety of sub-applications such as neurooncology, neurodegeneration, or stroke. Finally, we used the radiomics approach for feature extraction, which is more transparent and easier to understand compared to deep learning approaches. However, all steps must be carried out manually and hence may have unfavorable time/labor profiles. Therefore, the next step is to implement deep neural networks for brain tumor segmentation and feature extraction. In the next step, the application of innovative AI frameworks [90,91] would then be conceivable to uncover deep interrelationships between gene expression, neuropathology, and clinical data.

## 5. Conclusions

In summary, innovative physio-metabolic imaging methods provide valuable functional and metabolic insights into tumor pathophysiology and enable a reliable preoperative classification of the *IDH* gene status of gliomas even with a smaller amount of data. However, this advantage is partly lost in multisite clinical AI applications compared to anatomical or clinically advanced MRI due to varying data acquisition techniques. The large amounts of anatomical MRI data in public databases permit exploration of more complex AI algorithms more suitable for this purpose. A two-step protocol capable of counterbalancing these disadvantages and variabilities in experimental imaging techniques could be a foundation model trained in advance through self-supervised learning for brain imaging in general and then adapted for specific tasks using much smaller amounts of data.

## Figures and Tables

**Figure 1 cancers-16-01102-f001:**
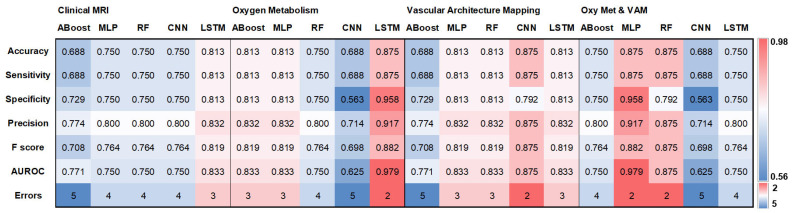
Overview of the results for the testing with the independent internal patient cohort. Heat maps depicting (top down) accuracy, sensitivity, specificity, precision, F-score, AUROC, and classification errors for the traditional ML algorithms adaptive boosting (ABoost), multilayer perceptron (MLP), and random forest (RF), as well as for the DL algorithms CNN and LSTM trained with the for the four MRI data sets (clinical MRI, MRI-based oxygen metabolism, MRI-based vascular architecture mapping (VAM), and the combination of oxygen metabolism and VAM). The color codes are listed on the far right.

**Figure 2 cancers-16-01102-f002:**
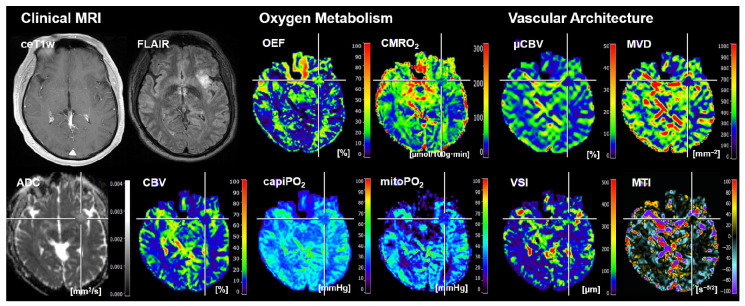
Representative case of the independent internal patient cohort shows a 54-year-old male patient who suffered from an *IDH*-mutated oligodendroglioma WHO grade 2 that was correctly classified by all algorithms. Clinical MRI data included contrast-enhanced (CE) T1w MRI, FLAIR MRI, and the quantitative maps of the apparent diffusion coefficient (ADC) and cerebral blood volume (CBV). MRI biomarker maps for oxygen metabolism included the oxygen extraction fraction (OEF), cerebral metabolic rate of oxygen (CMRO_2_), capillary oxygen tension (capiPO_2_), and mitochondrial oxygen tension (mitoPO_2_), respectively. MRI biomarker maps for microvascular architecture and neovascularization activity included the microvascular perfusion (µCBV), microvascular density (MVD), microvascular diameter (aka vessel size index, VSI), and microvascular type (MTI). The color codes are presented on the right-hand part of the images.

**Figure 3 cancers-16-01102-f003:**
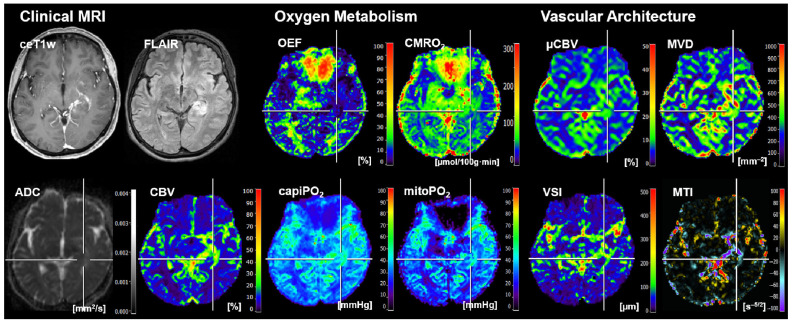
Representative case of the independent internal patient cohort shows a 60-year-old female patient who suffered from an *IDH*-wildtype astrocytoma WHO grade 3 that was correctly classified by all algorithms. Clinical MRI data included contrast-enhanced (CE) T1w MRI, FLAIR MRI, and the quantitative maps of the apparent diffusion coefficient (ADC) and cerebral blood volume (CBV). MRI biomarker maps for oxygen metabolism included the oxygen extraction fraction (OEF), cerebral metabolic rate of oxygen (CMRO_2_), capillary oxygen tension (capiPO_2_), and mitochondrial oxygen tension (mitoPO_2_), respectively. MRI biomarker maps for microvascular architecture and neovascularization activity included the microvascular perfusion (µCBV), microvascular density (MVD), microvascular diameter (aka vessel size index, VSI), and microvascular type (MTI). The color codes are presented on the right-hand part of the images.

**Figure 4 cancers-16-01102-f004:**
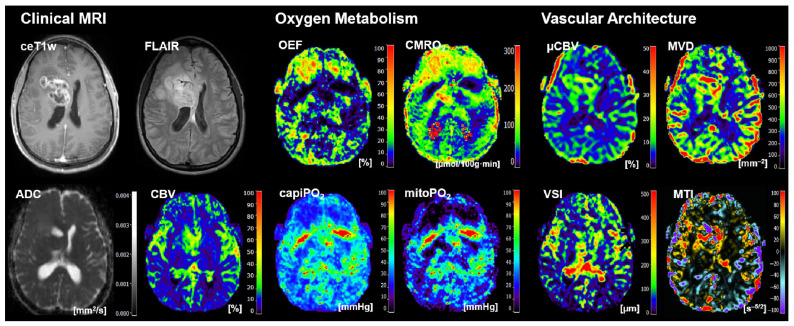
Case of a 36-year-old female patient who suffered from an *IDH*-wildtype glioblastoma WHO grade 4 that was misclassified by all algorithms. The reason for this was significant movement artifacts in the perfusion and relaxometry data. Clinical MRI data included contrast-enhanced (CE) T1w MRI, FLAIR MRI, and the quantitative maps of the apparent diffusion coefficient (ADC) and cerebral blood volume (CBV). MRI biomarker maps for oxygen metabolism included the oxygen extraction fraction (OEF), cerebral metabolic rate of oxygen (CMRO_2_), capillary oxygen tension (capiPO_2_), and mitochondrial oxygen tension (mitoPO_2_), respectively. MRI biomarker maps for microvascular architecture and neovascularization activity included the microvascular perfusion (µCBV), microvascular density (MVD), microvascular diameter (aka vessel size index, VSI), and microvascular type (MTI). The color codes are presented on the right-hand part of the images.

**Figure 5 cancers-16-01102-f005:**
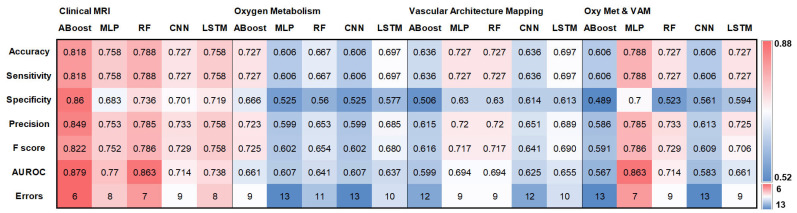
Overview of the results for the testing with the independent external patient cohort. Heat maps depicting (top down) accuracy, sensitivity, specificity, precision, F-score, AUROC, and classification errors for the traditional ML algorithms adaptive boosting (ABoost), multilayer perceptron (MLP), and random forest (RF), as well as for the DL algorithms CNN and LSTM trained with the for the four MRI data sets (clinical MRI, MRI-based oxygen metabolism, MRI-based vascular architecture mapping (VAM), and the combination of oxygen metabolism and VAM). The color codes are listed on the far right.

**Figure 6 cancers-16-01102-f006:**
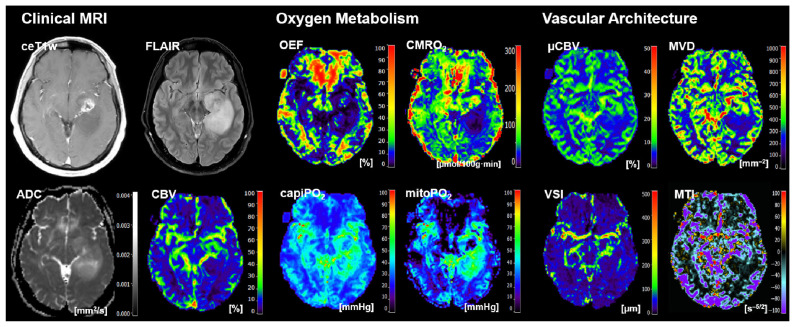
Case of the independent external patient cohort: A 45-year-old male patient who suffered from an *IDH*-wildtype glioblastoma WHO grade 4 that was misclassified by all algorithms. The reason for this was signal changes that are atypical for a glioblastoma, such as no contrast enhancement in large parts of the tumor, low hyperperfusion and neovascularization, and high oxygen tension. Clinical MRI data included contrast-enhanced (CE) T1w MRI, FLAIR MRI, and the quantitative maps of the apparent diffusion coefficient (ADC) and cerebral blood volume (CBV). MRI biomarker maps for oxygen metabolism included the oxygen extraction fraction (OEF), cerebral metabolic rate of oxygen (CMRO_2_), capillary oxygen tension (capiPO_2_), and mitochondrial oxygen tension (mitoPO_2_), respectively. MRI biomarker maps for microvascular architecture and neovascularization activity included the microvascular perfusion (µCBV), microvascular density (MVD), microvascular diameter (aka vessel size index, VSI), and microvascular type (MTI). The color codes are presented on the right-hand part of the images.

**Figure 7 cancers-16-01102-f007:**
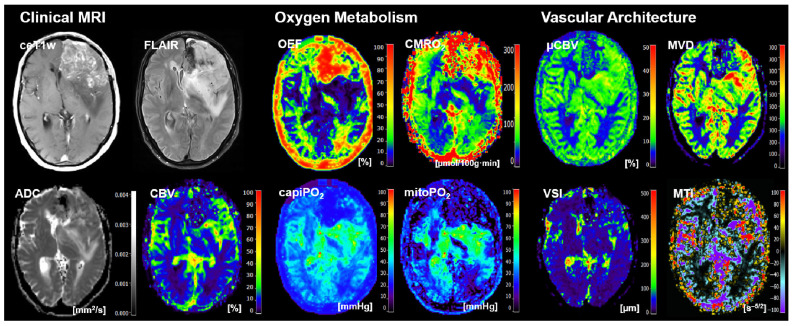
Case of the independent external patient cohort: A 67-year-old male patient who suffered from an *IDH*-mutated oligodendroglioma WHO grade 3 that was misclassified by all algorithms. The reason for this was susceptibility artifacts due to bleeding in the tumor. Clinical MRI data included contrast-enhanced (CE) T1w MRI, FLAIR MRI, and the quantitative maps of the apparent diffusion coefficient (ADC) and cerebral blood volume (CBV). MRI biomarker maps for oxygen metabolism included the oxygen extraction fraction (OEF), cerebral metabolic rate of oxygen (CMRO_2_), capillary oxygen tension (capiPO_2_), and mitochondrial oxygen tension (mitoPO_2_), respectively. MRI biomarker maps for microvascular architecture and neovascularization activity included the microvascular perfusion (µCBV), microvascular density (MVD), microvascular diameter (aka vessel size index, VSI), and microvascular type (MTI). The color codes are presented on the right-hand part of the images.

**Table 1 cancers-16-01102-t001:** Accuracy values for validation of the trained models.

	ABoost	MLP	RF	CNN	LSTM
Clinical MRI	0.858	0.866	0.907	0.891	0.94
Oxygen Metabolism	0.87	0.907	0.902	0.855	0.88
Vascular Architecture Mapping	0.85	0.902	0.907	0.891	0.88
Oxy Met & VAM	0.829	0.902	0.898	0.873	0.86

Oxy Met = Oxygen metabolism; VAM = vascular architecture mapping; ABoost = adaptive boosting; MLP = multilayer perceptron; RF = random forest, CNN = convolutional neural network; LSTM = long short-term memory.

## Data Availability

Data available on request due to privacy and ethical restrictions.

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
