# Peer review of "Machine Learning-Based Prediction of Glioma IDH Gene Mutation Status Using Physio-Metabolic MRI of Oxygen Metabolism and Neovascularization (A Bicenter Study)"

_cancers, 2024, doi:10.3390/cancers16061102_

Round 1

Reviewer 1 Report

Comments and Suggestions for Authors

In this paper the authors present an effort for IDH mutation status prediction from multi parametric MRI (anatomical,qBOLD and DWI) in gliomas. Methods, Results and Discussion sections are very clear to me, I only have some minor/constructive comments.

1) Since there is an Institutional Review Board Statement the 2.1 section needs to be removed.

2) 3.1. Patient Characteristics of the Cohorts should be placed under the patient section. (It could be nicer if patient statistics could be tabularized per center as in supl. table 1.)

3) In section 2.4 the authors should consider adding the b values used for the calculation of ADC (i.e b = 0, 1000 s/mm2). Also the choice of those values should be discussed in the discussion as a limitation since different high b values  (i.e 1000, or 1500) should affect the ADC calculation. This limitation can be ignored if more b values are in present and instead of the linear aproximation of ADC in  eq1, non linear least squares are applied to fit the b-value decay curve into the DWI data. (also more quantitative biomarkers can be calculated such as IVIM parameters).

4) Lastly, to provide a more sufficient background in the Introduction section, the authors are recommended to include 2 more recent and relevant references  in IDH mutation status prediction.

https://doi.org/10.3390/cancers13163965

and

https://doi.org/10.3389/fneur.2023.1249452

Author Response

Response to Review Report of Reviewer 1

Comment 1) Since there is an Institutional Review Board Statement the 2.1 section needs to be removed.

Response 1:  We want to thank the reviewer for this comment. We agree with the reviewer and removed Section 2.1. Ethics.

Comment 2) 3.1. Patient Characteristics of the Cohorts should be placed under the patient section. (It could be nicer if patient statistics could be tabularized per center as in supl. table 1.)

Response 2:  We thank the reviewer for this comment. We moved the patient characteristics of the cohorts to the Patients section 2.1 and summarized the details in a new Supplementary Table 1.

Comment 3) In section 2.4 the authors should consider adding the b values used for the calculation of ADC (i.e b = 0, 1000 s/mm2). Also the choice of those values should be discussed in the discussion as a limitation since different high b values  (i.e 1000, or 1500) should affect the ADC calculation. This limitation can be ignored if more b values are in present and instead of the linear aproximation of ADC in  eq1, non linear least squares are applied to fit the b-value decay curve into the DWI data. (also more quantitative biomarkers can be calculated such as IVIM parameters).

Response 3:  The reviewer is absolutely right. We want to thank the reviewer for this important comment. We described the calculation of the ADC in more detail and discussed the limitations in the discussion section.

Comment 4) Lastly, to provide a more sufficient background in the Introduction section, the authors are recommended to include 2 more recent and relevant references  in IDH mutation status prediction. https://doi.org/10.3390/cancers13163965 and https://doi.org/10.3389/fneur.2023.1249452

Response 4:  We would like to thank the reviewer for pointing out the very interesting studies. We discussed the two studies in the discussion section.

Reviewer 2 Report

Comments and Suggestions for Authors

Stadlbauer and colleagues report an interesting analysis that aims to compare the ability of clinical MRI versus physio-metabolic MRI to be employed for the prediction of IDH wild type or mutant glioblastoma. The models are developed using a cohort of 182 patients, 166 for training and 16 for testing; then, validated on an external cohort of 33 patients. The classes in these cohorts are very unbalanced (135 IDHwt and 47 IDHmut in the first cohort, and 21 IDHwt and 12 IDHmut in the external validation cohort). This is an issue that requires careful assessment and special treatment when developing a predictive model.

I have a major concern on the results on model performance reported in Figures 1 and 5, specifically that the reported values should not be possible. In particular, the values for accuracy, sensitivity, specificity and precision appear to be incorrect. As an example, let's consider the results in Figure 1 which refer to the predictions on the test set of 16 patients (12 IDHwt and 4 IDHmut). Figure 1 , section 'Clinical MRI' and column 'ABoost': sensitivity (the proportion of true positives) = 0.688 --> 0.688 * 12 IDHwt cases = 8.256, indicating that 8.256 patients that were IDHwt were correctly classified as IDHwt. Similarly, specificity (the proportion of true negatives) = 0.688 --> 0.688 * 4 IDHmut cases = 2.752, indicating that 2.752 patients that were IDHmut were correctly classified as IDHmut. 

Models output a single prediction for each patients, therefore it is not possible to say that, for example: "2.752 patients that were IDHmut were correctly classified as IDHmut". Either 3 patients or 2 patients that were IDHmut were correctly classified as IDHmut. In other words, when calculating out (as shown) all the percents reported for accuracy, sensitivity, specificity and precision for each model, the resulting number of patients predicted to be part of one class or another should be INTEGERS in all cases.

Can the authors explain how were the results reported in Figures 1 and 5 obtained? Importantly, all confusion matrices for all reported models should be included in the paper (20 confusion matrices for Figure 1, and 20 confusion matrices for Figure 5), which would greatly help in supporting the reported results.

Other major concerns are as follows:

- the authors do not provide enough details to assess whether the issue of unbalanced classes is appropriately taken into account; do the models overestimate the probability to belong to the minority class (IDH mutant)? This can be best assessed by extracting the predicted probabilities from the output layer of each model, and plotting  calibration curves.

- additional details should be provided on model optimization. For each model, which hyperparameters were optimized? What were the ranges of values for each hyperparameter? Was the model architecture optimized for the deep learning models? Were combinations of different number of layers and different number of neurons within layers tested? How was the best set of hyperparameters selected? How was the best model architecture selected?

Minor concerns

- Please include a legend for Supplementary Figure 1, specifying all acronyms and their meaning.

- please include schematic diagrams depicting the procedure  for model optimization (one for each model). The diagram should specify parameters optimized, rounds of optimization, method to pick best set of hyperparameters

- The background section regarding the applications of deep learning models to biological and medical problems should be expanded and recent papers presenting innovative applications should be cited, including:

- DeepOmicsAE: Representing Signaling Modules in Alzheimer's Disease with Deep Learning Analysis of Proteomics, Metabolomics, and Clinical Data, Panizza E. J Vis Exp. 2023 Dec 15:(202).  

- Unified AI framework to uncover deep interrelationships between gene expression and Alzheimer's disease neuropathologies. Beebe-Wang N et al., Nat Commun. 2021 Sep 10;12(1):5369.  

Author Response

Response to Review Report of Reviewer 2

Comment 1. I have a major concern on the results on model performance reported in Figures 1 and 5, specifically that the reported values should not be possible. In particular, the values for accuracy, sensitivity, specificity and precision appear to be incorrect. As an example, let's consider the results in Figure 1 which refer to the predictions on the test set of 16 patients (12 IDHwt and 4 IDHmut). Figure 1 , section 'Clinical MRI' and column 'ABoost': sensitivity (the proportion of true positives) = 0.688 --> 0.688 * 12 IDHwt cases = 8.256, indicating that 8.256 patients that were IDHwt were correctly classified as IDHwt. Similarly, specificity (the proportion of true negatives) = 0.688 --> 0.688 * 4 IDHmut cases = 2.752, indicating that 2.752 patients that were IDHmut were correctly classified as IDHmut. 

Models output a single prediction for each patients, therefore it is not possible to say that, for example: "2.752 patients that were IDHmut were correctly classified as IDHmut". Either 3 patients or 2 patients that were IDHmut were correctly classified as IDHmut. In other words, when calculating out (as shown) all the percents reported for accuracy, sensitivity, specificity and precision for each model, the resulting number of patients predicted to be part of one class or another should be INTEGERS in all cases.

Can the authors explain how were the results reported in Figures 1 and 5 obtained? Importantly, all confusion matrices for all reported models should be included in the paper (20 confusion matrices for Figure 1, and 20 confusion matrices for Figure 5), which would greatly help in supporting the reported results.

Response 1:  We want to thank the reviewer for this comment. The performance values given in Figures 1 and 5 refer to the entire test cohort, i.e. 16 patients in Figure 1 and 33 patients in Figure 5. Our data for both independent internal and independent external testing were is imbalanced data set. In order to assign larger importance to the class with more patients (IDHwt) we used weighted averaging. This approach takes into account the balance of classes by weighing each class based on its representation in the dataset. We compute the performance values as a weighted mean of the performance values in individual classes (IDHwt and IDHmut). We included this information into the Methods section. There were also a few typos that were corrected.

We included the confusion matrices into a new Supplementary Figure 2 in the Supplementary Results section.

Comment 2. the authors do not provide enough details to assess whether the issue of unbalanced classes is appropriately taken into account; do the models overestimate the probability to belong to the minority class (IDH mutant)? This can be best assessed by extracting the predicted probabilities from the output layer of each model, and plotting  calibration curves.

Response 2:  We thank the reviewer for this comment. We refer to our response to comment 1.

Comment 3. additional details should be provided on model optimization. For each model, which hyperparameters were optimized? What were the ranges of values for each hyperparameter? Was the model architecture optimized for the deep learning models? Were combinations of different number of layers and different number of neurons within layers tested? How was the best set of hyperparameters selected? How was the best model architecture selected?

Response 3:  We thank the reviewer for this comment. We agree with the reviewer that model optimization and hyperparameter optimization is an important part of model development. However, we would like to point out that the aim of our study, according to the topic of the Special Issue, was innovative imaging methods of brain tumors. In our case, a clinical application of physiological MRI in combination of established ML and DL methods. In our opinion, a detailed description of the individual hyperparameter optimization goes beyond the scope of our study. The parameters suggested by the WEKA software were essentially used for the ML models. For the DL models, all specified parameters have been optimized. We have included a table with the parameters in the Supplementary Methods.

Comment 4. Please include a legend for Supplementary Figure 1, specifying all acronyms and their meaning.

Response 4:  We thank the reviewer for this comment. We included a legend for Supplementary Figure 1, specifying all acronyms and their meaning.

Comment 5. please include schematic diagrams depicting the procedure  for model optimization (one for each model). The diagram should specify parameters optimized, rounds of optimization, method to pick best set of hyperparameters

Response 5:  We thank the reviewer for this comment. We refer to our response to comment 3.

Comment 6. The background section regarding the applications of deep learning models to biological and medical problems should be expanded and recent papers presenting innovative applications should be cited, including:

- DeepOmicsAE: Representing Signaling Modules in Alzheimer's Disease with Deep Learning Analysis of Proteomics, Metabolomics, and Clinical Data, Panizza E. J Vis Exp. 2023 Dec 15:(202).  

- Unified AI framework to uncover deep interrelationships between gene expression and Alzheimer's disease neuropathologies. Beebe-Wang N et al., Nat Commun. 2021 Sep 10;12(1):5369.  

Response 6:  We would like to thank the reviewer for pointing out the very interesting studies. We have cited the two papers in the discussion section.

Round 2

Reviewer 2 Report

Comments and Suggestions for Authors

I thank the authors for the revision of the paper. A few outstanding issues remain that needs to be addressed prior to publication:

Reponse 1: The authors explain that they use a weighted mean approach to calculate the reported performance metrics, however the values reported in Figures 1 and 5 are incorrect and must be amended prior to publication. For example, for Figure 1, Clinical MRI, ABoost the unweighted specificity for wt = 0.428 and the unweighted specificity for mut = 0.889. The weighted specificity is: ((0.428*12)+(0.889*4)) / (12+4) = 0.544 (instead of the value of 0.688 reported from the authors). The mistake in the reported results may stem from the application of the 'weighted' method in the 'recall_score' function from the Python package 'sklearn'. When calculating the specificity with this python function a warning is returned, and the output is an incorrect value matching the one reported by the authors. The warning is meant to communicate that that the 'recall_score' function is designed to calculate sensitivity, and that the other measures will be inaccurate.

The authors should implement manual calculations of the weighted performance values as outlined above and correct the results reported in Figure 1 and 5.

Response 3: Please provide in the Methods details on how the WEKA software was used, which settings were selected, which methods applied, etc

Response 6: The studies have not been included in the list of referenced papers. Please make sure that you have added the citations.

Author Response

Response to Review Report of Reviewer 2

Comment 1: The authors explain that they use a weighted mean approach to calculate the reported performance metrics, however the values reported in Figures 1 and 5 are incorrect and must be amended prior to publication. For example, for Figure 1, Clinical MRI, ABoost the unweighted specificity for wt = 0.428 and the unweighted specificity for mut = 0.889. The weighted specificity is: ((0.428*12)+(0.889*4)) / (12+4) = 0.544 (instead of the value of 0.688 reported from the authors). The mistake in the reported results may stem from the application of the 'weighted' method in the 'recall_score' function from the Python package 'sklearn'. When calculating the specificity with this python function a warning is returned, and the output is an incorrect value matching the one reported by the authors. The warning is meant to communicate that that the 'recall_score' function is designed to calculate sensitivity, and that the other measures will be inaccurate.

The authors should implement manual calculations of the weighted performance values as outlined above and correct the results reported in Figure 1 and 5.

Response 1: We want to thank the reviewer for this comment. We calculated the parameters in the traditional manner from the confusion matrix using the generally accepted and used formulas for the parameters (https://en.wikipedia.org/wiki/Confusion_matrix). We didn't use any Python packages or a recall_score function. In other words: We have actually always carried out manual calculations. This information is included into the Supplementary Materials as Supplementary Figure 2.

Here the values for the example Clinical MRI, ABoost:

All values are calculated using these equations. In our opinion they are correct.

Comment 3: Please provide in the Methods details on how the WEKA software was used, which settings were selected, which methods applied, etc

Response 6: We thank the reviewer for this comment. We included details how the WEKA software was used in the Methods section.

Comment 6: The studies have not been included in the list of referenced papers. Please make sure that you have added the citations.

Response 6: We refer to the references numbered 90 and 91.

Round 3

Reviewer 2 Report

Comments and Suggestions for Authors

All my concerns were appropriately addressed. I thank the authors for the revision work.